# S-Methyl-L-Ergothioneine to L-Ergothioneine Ratio in Urine Is a Marker of Cystine Lithiasis in a Cystinuria Mouse Model

**DOI:** 10.3390/antiox10091424

**Published:** 2021-09-07

**Authors:** Miguel López de Heredia, Lourdes Muñoz, Ciriaco Carru, Salvatore Sotgia, Angelo Zinellu, Carmen Serra, Amadeu Llebaria, Yukio Kato, Virginia Nunes

**Affiliations:** 1Human Molecular Genetics Laboratory, Gene, Disease and Therapy Program, IDIBELL, L’Hospitalet de Llobregat, 08908 Barcelona, Spain; 2Centro de Investigación Biomédica en Red de Enfermedades Raras (CIBERER)-CB06/07/0069, Instituto de Salud Carlos III, 28029 Madrid, Spain; 3SIMChem, Institute for Advanced Chemistry of Catalonia (IQAC-CSIC), 08034 Barcelona, Spain; lourdes.munoz@iqac.csic.es (L.M.); cscqbi@cid.csic.es (C.S.); amadeu.llebaria@iqac.csic.es (A.L.); 4Department of Biomedical Sciences, University of Sassari, 07100 Sassari, Italy; carru@uniss.it (C.C.); ssotgia@uniss.it (S.S.); azinellu@uniss.it (A.Z.); 5MCS, Laboratory of Medicinal Chemistry, Institute for Advanced Chemistry of Catalonia (IQAC-CSIC), 08034 Barcelona, Spain; 6Faculty of Pharmacy, Kanazawa University, Kanazawa 920-1192, Japan; ykato@p.kanazawa-u.ac.jp; 7Genetics Section, Physiological Sciences Department, Health Sciences and Medicine Faculty, University of Barcelona, 08907 Barcelona, Spain

**Keywords:** L-ergothioneine, S-methyl-L-ergothioneine, cystine lithiasis biomarker, cystinuria, *Slc7a9* KO mice

## Abstract

Cystinuria, a rare inherited aminoaciduria condition, is characterized by the hyperexcretion of cystine, ornithine, lysine, and arginine. Its main clinical manifestation is cystine stone formation in the urinary tract, being responsible for 1–2% total and 6–8% pediatric lithiasis. Cystinuria patients suffer from recurrent lithiasic episodes that might end in surgical interventions, progressive renal functional deterioration, and kidney loss. Cystinuria is monitored for the presence of urinary cystine stones by crystalluria, imaging techniques or urinary cystine capacity; all with limited predicting capabilities. We analyzed blood and urine levels of the natural antioxidant L-ergothioneine in a Type B cystinuria mouse model, and urine levels of its metabolic product S-methyl-L-ergothioneine, in both male and female mice at two different ages and with different lithiasic phenotype. Urinary levels of S-methyl-L-ergothioneine showed differences related to age, gender and lithiasic phenotype. Once normalized by L-ergothioneine to account for interindividual differences, the S-methyl-L-ergothioneine to L-ergothioneine urinary ratio discriminated between cystine lithiasic phenotypes. Urine S-methyl-L-ergothioneine to L-ergothioneine ratio could be easily determined in urine and, as being capable of discriminating between cystine lithiasis phenotypes, it could be used as a lithiasis biomarker in cystinuria patient management.

## 1. Introduction

L-Ergothioneine (L-Erg) is a natural thiohistidine [1] absorbed from the diet in mammals and synthesized by non-yeast-like fungi, actinobacteria [2], methylotrophs [3] and cyanobacteria [4]. L-Erg is widely distributed among tissues, being more abundant in erythrocytes, liver, seminal fluid, bone marrow, eye lens, cornea and kidneys [5,6,7]. L-Erg has been shown to accumulate in injured tissues like joints in collagen-induced arthritis [8], inflamed intestinal mucosa of patients with Crohn’s disease [9], and liver damage [10,11]. This has pointed to the hypothesis that it might function in parallel to known adaptative antioxidants [2]. L-Erg is specifically transported by OCTN1 (*Slc22a4*) [12,13] and the OCTN1 knock out mice (*Slc22a4^−/−^*) shows complete absence of L-Erg in tissues [14]. 

Cystinuria, a rare aminoaciduria condition (OMIM#220100 or OMIM#600918) with an estimated prevalence of 1:7000 newborns is caused by mutations in either *SLC3A1* or *SLC7A9* genes, which encode rBAT and b^0,+^ AT, respectively; the two subunits of the cystine and dibasic b^0,+^ amino acid transporter of renal and intestinal epithelial cells. The disease is characterized by the urine hyperexcretion of cystine and dibasic amino acids (lysine, ornithine, and arginine). The main clinical symptom is the generation of cystine calculi in the urinary tract due to cystine low solubility at urine physiological pH. Cystine lithiasis represents 1–2% total and 6–8% pediatric renal lithiasis [15], and is monitored for the presence of urinary cystine stones by crystalluria [16], imaging techniques or urinary cystine capacity [17]; all with limited predicting capabilities. Affected patients suffer from recurrent lithiasic episodes that can lead to progressive renal functional deterioration [18,19] and often lead to renal failure limiting patient’s quality of life. Lithiasic episodes may cluster between periods without symptomatic stone disease [20], and cystine stones are often removed by surgical interventions. The risk of CKD among patients with hereditary stone diseases like cystinuria is higher than patients with other stone disease [18]. Stone recurrence, repeated surgical procedures to remove unsolved stones, obstructive uropathy, urinary tract infections, nephrotoxic treatments and/obstructions of Bellini ducts by cystine crystals may be behind the observed prevalence of CKD in cystinuria patients [18,21]. It has been recently shown L-Erg levels are lower in mice and patients with CKD [22]. As no reliable biomarker of stone formation in cystinuria exists [16], new methods for predicting the lithiasic episodes, reliable and easy to set up in clinical diagnostic laboratories, are needed.

Mouse models of cystinuria resemble human cystinuria to a large extent (reviewed extensively in [21]) and the cystinuria Type B mouse model (*Slc7a9^−/−^*), in particular: (i) forms cystine stones from early adulthood in both sexes; (ii) cystine, arginine, lysine and ornithine excretion profiles are similar to those found in Type B cystinuria patients [23]; and, (iii) has been shown to reproduce the effects of a currently used cystinuria treatment and be a good model for antilithiasic pharmacological studies [24].

In the present study, we analyzed blood and 24h urine concentrations and excretion of L-Erg and one of its subproducts, S-methyl-L-ergothioneine (S-Met-L-Erg) in the cystinuria Type B mouse model (*Slc7a9^−/−^*) [23] in both sexes at 3 and 6 months of age, evaluate their putative role as a cystine lithiasis biomarker, and the biomarker capacity as a cystine lithiasis predictor.

## 2. Materials and Methods

### 2.1. Mice Care

Mice were maintained in a 12 h light-dark cycle in a humidity and temperature-controlled room. Animals were housed in sterile cages with free access to food (Teklad Global 14% Protein Diet, Harlan Laboratories, Madison, WI, USA) and water.

### 2.2. Knocking Out Slc22a4 in the Type B Cystinuric Mouse Model (Slc7a9^−/−^)

Single loss-of-function mouse models for *Slc7a9^−/−^* (NCBI gene ID: 30962, location: NC_000073.6) [23] and *Slc22a4^−/−^* (NCBI gene ID: 30805, location: NC_000077.6) [14], both in a pure C57BL/6 J genetic background were crossed to obtain double heterozygous mice, which were backcrossed to get all expected genotypes. Only those mice with a *Slc7a9^−/−^* genotype and with all *Slc22a4* genotype combinations, including the double KO *Slc7a9^−/−^ Slc22a4^−/−^* (dKO), were further used. For genotyping analyses, genomic DNA was isolated from tail tissue. *Slc22a4^−/−^* genotype was confirmed by PCR (30 cycles at 60 °C annealing temperature), based on a 3′-primer strategy (F: 5′-GGGTGTGGTCCAGAGGACT-3′; R WT-specific: 5′-TAGTTGCCAGCCATCTGTTG-3′; R KO-specific: 5′-GACTGACATACC ATTGAAGC-3′) allowing to distinguish genotypes by generating 255 bp and 313 bp fragments from the WT and KO alleles, respectively. For *Slc7a9^−/−^*, genotype was confirmed by PCR (30 cycles at 60 °C annealing temperature), based on a 3′-primer strategy (F: 5′-GCATTCGCCACAGGCTCTTC-3′; R-WT: 5′-CTGTGTTGGCCAGCACAGAC-3′, R KO-specific: 5′-CGCAGCGCATCGCCTTCTAT-3′), allowing to distinguish genotypes by generating 452 bp and 311 bp fragments from the WT and KO alleles, respectively.

### 2.3. Cystine Calculi Detection by X-ray In Vivo Imaging

At 3 and 6 months of age, isoflurane anesthetized mice were subjected to X-ray imaging for lithiasis detection in an IVIS Lumina XR Series III (Caliper Lifescience—Vertex Techniques, Hopkinton, MA, USA) following the manufacturer’s imaging parameters with a calibration curve of cystine stones of known weights. Mice with a cystine stone present at 3-months of age were catalogued as Early Stone Formers (ESF) and those with a cystine lithiasis onset between 3 and 6-months of age were catalogued as Late Stone Formers (LSF). Those without a lithiasic phenotype were considered non-stone formers (NSF).

### 2.4. Sample Collection

Mice were individually housed in metabolic cages for 4 days with the first day as an adaptation period. Mice weight, water and food intake, and excreted urine were monitored daily. 24 h urine samples were collected and kept at −80 °C until further analysis with 50 µL 10% thymol in isopropanol as preservative. On the last day, blood was obtained by intracardiac puncture with EDTA coated syringes and transferred into Microvette EDTA-tubes (Sarstedt, Nümbrecht, Germany) on isoflurane anesthetized animals and centrifuged at 3000 rpm for 10 min and 4 °C in a minifuge after 10 min incubation at room temperature. Plasma was then separated into a new tube and kept on ice. Plasma absorbance at 414 nm was then determined to quantify hemolysis with a NanoDrop ND-1000 spectrophotometer (Thermo Fisher Scientific, Waltham, MA, USA) and only those with OD < 0.2 were considered for further analysis. Blood and plasma samples were stored at −80 °C and centrifuged erythrocytes (RBCs) were also collected and stored at −80 °C.

### 2.5. L-Erg and S-Met-L-Erg Determination 

L-Erg in blood, plasma and RBCs was measured as described by Sotgia et al. [25,26] while plasma creatinine as described by Zinellu et al. [27]. 

For L-Erg and S-Met-L-Erg analysis in urine, thawed urine samples removed from debris were 1:10 diluted in Milli-Q water supplemented with 3×-deuterated L-Erg and S-Met-L-Erg (50 ng/mL final concentration each). The L-Erg, S-Met-L-Erg and their respective 3×-deuterated isoforms used as standards where in-house synthesized following a previously described route [28]. Both, diluted samples and standards were then filtered through eXtremeFV PVDF 0.2 µm filter vials (Thomson Instrument Company, Oceansite, CA, USA), and L-Erg and S-Met-L-Erg quantified from 20 μL by LC-MS/MS using an LPG-3400SD LC System (Dionex, Sunnyvale, CA, USA) coupled to an LTQ-XL ESI tandem mass spectrometer (Thermo, Burlington, MA, USA). Samples and standards were kept at 15 °C in the autosampler and were injected into a ZORBAX Eclipse Plus C18 (3.5 μm, 75 × 4.6 mm; Agilent, Sta. Clara, CA, USA) maintained at 35 °C. Solvent A was 0.05% formic acid in ultrapure water, and solvent B was acetonitrile in 0.05% formic acid. Chromatography was carried out at a flow rate of 0.9 mL/min under isocratic conditions (99%A:1%B) for 3 min. Mass spectrometry was carried out under positive ion, electrospray ionization mode, using multiple reaction monitoring (MMR) for quantification of specific target ions. Source voltage was set at 3.0 kV, and capillary temperature was kept at 375 °C. Nitrogen sheath gas flow was 90 (a.u.), auxiliary gas flow 10 (a.u.) and sweep gas flow was 6 (a.u.). Alphagaz 2 helium (Air Liquide, Paris, France) was used as collision gas. Precursor to product ion transitions for each compound were as follows: L-Erg: 230.0 → 186.0; d3-L-Erg: 233.0 → 189.0; S-Met-L-Erg: 244.0 → 200.0; d3-S- Met-L-Erg: 247.0 → 203.0. In all cases, isolation width (*m*/*z*) and CID collision energies were 2.0 and 20%, respectively. Creatinine concentrations in 24 h urine samples were determined with Creatinine Assay Kit (Sigma, Sant Louis, MO, USA) as indicated by manufacturer after filtering through 10 kDa MWCO spin filters (Amicon Ultra 0.5 mL, Millipore, Burlington, MA, USA).

### 2.6. Statistical and Prediction Analysis

RStudio and ggplot2, xlsx, dplyr, ggpubr, plotROC, stringr, tidyr, ggsignif, and plyr R packages under R version 4.0.3 were used for data analysis and figure preparation. Mann-Whitney statistical test was used for statistical analysis and statistical significance was considered if *p* ≤ 0.05.

## 3. Results

### 3.1. L-Erg and S-Met-L-Erg in Male Mice Blood and Urine

We first analyzed L-Erg content in blood and urine, S-Met-L-Erg concentration in urine and L-Erg and S-Met-L-Erg excretion of 3 months-old WT and *Slc7a9^−/−^* male mice (shown in Figure 1, 3 months). No statistically significant differences were seen although a lower content of S-Met-L-Erg urine concentration and excretion in *Slc7a9^−/−^* male mice compared to WT mice have been detected at this age.

Age related differences in RBCs’ L-Erg concentration were reported in humans [29] and in rats [30]. To account for this results in mice, we analyzed as before blood L-Erg levels and urine L-Erg and S-Met-L-Erg levels in male mice at 6 months of age (shown in Figure 1, 6 months). We observed statistically 33% and 41% significant reductions at 6-months of age in S-Met-L-Erg concentration and excretion, respectively, in *Slc7a9**^−/−^* male mice compared to WT mice (shown in Figure 1B,C, 6 months).

When both ages were compared, only L-Erg levels in RBCs and urine were significatively 2-fold higher at 6 months of age in both WT and *Slc7a9**^−/−^* male mice (shown in Figure 1A, RBC and urine).

### 3.2. L-Erg and S-Met-L-Erg in Female Mice Blood and Urine

Gender related L-Erg differences in RBC levels have also been described in female rats [30]. We then investigated as above L-Erg concentration in blood, plasma and RBCs, and L-Erg and S-Met-L-Erg concentration and excretion in urine in 3- and 6-months-old WT and *Slc7a9^−/−^* female mice (shown in Figure 2). 

At 3 months of age (shown in Figure 2, 3 months), we only detected significant 20% and 31% lower S-Met-L-Erg urine concentration and excretion, respectively, in *Slc7a9^−/−^* female mice compared to WT (shown in Figure 2B,C, 3 months). At 6 months of age, reduced L-Erg levels were observed in plasma (31%) and RBCs (41%), and S-Met-L-Erg urine levels (27%) of *Slc7a9^−/−^* female mice compared to WT (shown in Figure 2, 6 months). We also compared both ages looking for age-related differences in female mice. We only observed an age-related significant decrease in L-Erg concentration in plasma in *Slc7a9^−/−^* female mice (shown in Figure 2A, plasma).

### 3.3. Sex-Related Differences in L-Erg and S-Met-L-Erg in Blood and Urine

We then looked for gender-related differences in WT and *Slc7a9^−/−^* (KO) female mice (shown in Figure 3), as a sex-related difference in L-Erg concentration in blood had been reported in rats [30]. 

Significative sex-related reduced L-Erg levels could be found in blood, plasma, RBCs, and urine of *Slc7a9^−/−^* female mice (43%, 36%, 48% and 30.8% lower, respectively) when compared to *Slc7a9^−/−^* male mice (shown in Figure 3A). Urine S-Met-L-Erg concentration in female mice showed significative differences in both WT and *Slc7a9^−/−^* mice at both tested ages, being about 40% lower at 3 months of age and about 50% lower at 6 months of age than corresponding male mice (shown in Figure 3B).

Except for WT mice at 3-months of age, L-Erg and S-Met-L-Erg urine excretions showed significant sex-related differences in both WT and *Slc7a9^−/−^* mice, being female mice L-Erg and S-Met-L-Erg excretion 41–72% lower than the excretion of male mice (shown in Figure 3C).

### 3.4. L-Erg and S-Met-L-Erg Urine Concentration in Lithiasic Mice

One of the hallmarks of cystinuria, which the Type B mouse model (*Slc7a9^−/−^*) reproduce [23], is the presence of cystine stones in cystinuria patients from both sexes. According to the age at which the stones were first detected, we classified lithiasic *Slc7a9^−/−^* mice as early stone formers (ESF) if the stone is first detected by X-ray analysis at 3 months of age, late stone formers (LSF) if first detected at 6 months of age, and non-stone formers (NSF) if not detected at this age.

Therefore, we analyzed, as above, L-Erg concentration in blood, plasma, RBCs and urine, S-Met-L-Erg levels in urine and L-Erg and S-Met-L-Erg excretions in stone and non-stone former (NSF) *Slc7a9**^−/−^* mice of both genders at 3 and 6 months of age (shown in Figure 4). L-Erg concentration in RBCs was significatively reduced in stone-former male mice at 6 months of age (shown in Figure 4A, RBC). Besides, L-Erg 24 h urine concentration was significatively higher in both ESF female and LSF male mice at 6 months of age and in ESF female mice at 3 months of age (shown in Figure 4A, urine). S-Met-L-Erg was lower in ESF females at 3 months of age and in stone former males at 6 months of age (shown in Figure 4B). L-Erg excretion was significant lower only in ESF males at 6 months of age and S-Met-L-Erg excretion in stone former mice from both sexes at both tested ages, 3 and 6 months of age (shown in Figure 4C).

As S-Met-L-Erg is a subproduct of L-Erg metabolism which levels are affected by those of L-Erg, we investigated if the urine concentration ratio between both compounds showed differences related to the lithiasic phenotype. As shown in Figure 5A, the ratio in stone former mice (SF) was a significatively 2- to 3-fold lower than in NSF *Slc7a9^−/−^* mice from both sexes at any age. This result strongly suggests that the ratio between the 24 h urine concentrations of S-Met-L-Erg and L-Erg could be used to differentiate the lithiasic phenotype in mice. We investigated this possibility by the means of a receiver operator characteristic (ROC) curve for 95 mice (54 lithiasic and 41 non-lithiasic) (shown in Figure 5B). The area under the ROC curve (AUC) of 84.9% (95% CI: 76.98–92.76%) and the negative predictive value of 78.04% and positive predictive value of 83.33% at the Youden’s cut-off value of 0.065 suggest the possibility of using this ratio as a lithiasic biomarker. Furthermore, the performance of the urinary ratio of S-Met-L-Erg to L-Erg was not severely affected by age (shown in Figure 5C) or sex (shown in Figure 5D).

### 3.5. OCTN1 as a Cystine Lithiasis Modulator

As L-Erg is transported by OCTN1 (*Slc22a4*) and as *Slc22a4^−/−^* mice lack L-Erg in the kidneys [14], we crossed the *Slc22a4^−/−^* mice with our cystinuria mice model (*Slc7a9^−/−^*) looking for differences in the rate of lithiasic mice related to OCTN1 loss (shown in Figure 6 and Table 1). We found a higher number of lithiasic female mice independently of *Slc22a4* genotype except at 48 weeks of age in *Slc7a9^−/−^ Slc22a4^+/−^* (Het, heterozygous for *Slc22a4*) where no age-related differences could be found. Generally, the rate of lithiasic mice was higher for the double KO (*Slc7a9^−/−^ Slc22a4^−/−^*) mice independently of sex, except at 48 weeks of age, in which no differences in the percentage of lithiasic mice could be related to the loss of *Scl22a4*. This result suggests that OCTN1 (*Slc22a4*) might modulate cystine lithiasis in mice by changing the cellular availability of the molecules it transports, especially L-Erg, as it is the main molecule transported known today.

To better understand this result, we analyzed the amount of L-Erg and S-Met-L-Erg in the 24 h urine of these mice at 3 months of age. L-Erg was detectable in 24 h urine to similar levels in *Slc7a9^−/−^ Slc22a4^+/+^* (WT for *Slc22a4*) mice when compared to wild-type mice (*Slc7a9^+/+^ Slc22a4^+/+^*), but the amount of S-Met-L-Erg was 50% lower (*p* = 0.0022) (shown in Table 2). Knocking out *Slc22a4* in the *Slc7a9^−/−^* mice produced a significative reduction of L-Erg levels in urine (*p* = 0.0382 for *Slc22a4^+/−^* and *p* = 0.0138 for *Slc22a4^−/−^*) and, although we could still detect S-Met-L-Erg, its 24 h urine concentration was below the quantification limits of the method used. As expected, double KO mice (*Slc7a9^−/−^ Slc22a4^−/−^*) showed lower concentration of L-Erg than *Slc7a9^−/−^ Slc22a4^+/−^* mice.

This result, similar to the observations made by Kato et al. when deleting OCTN1 and comparing L-Erg levels against WT mice [14], suggests that (i) another transporter might be involved in the absorption of L-Erg from the diet, (ii) OCTN1 is needed for L-Erg transport into cells, and (iii) S-Met-L-Erg is mainly produced intracellularly.

## 4. Discussion

L-Erg is particularly enriched in RBCs in comparison with plasma in WT mice; and L-Erg excretion is low, supporting its avid retention by the body [31], in agreement with previously published findings in humans [32,33,34]. But we have seen differences in either urine L-Erg concentration or excretion when comparing cystinuric mice with their WT counterparts.

In this sense, we have observed in 6-month-old cystinuric male mice a 15% reduction in L-Erg excretion without any change in 24 h urine concentration. L-Erg transporter, OCTN1, expression is upregulated in injured tissues to promote L-Erg uptake and help in reducing damage caused by oxidative stress [22]. Based on this, and due to the antioxidant and cytoprotective properties of L-Erg, this lower excretion might be an adaptive response against a lower antioxidant capacity of cystinuric kidneys, but the excretion was even lower for S-Met-L-Erg, a S-methyl derivative of L-Erg. No information on its biological source is available apart from being detectable in human blood and urine [32], and in many mouse tissues except spleen and heart [35]. The fact that its 24h urine concentration is low enough to avoid our quantification method in *Slc22a4^−/−^* mice suggests that it might be produced inside the cells perhaps in an unknown intracellular transmethylation reaction mediated by methyl-S-transferases. Based on our results, we hypothesized a reduced methylation capability of the kidney in cystinuria that could explain the observed reduced urine levels and excretion of S-Met-L-Erg in cystinuric mice.

The other known products of L-Erg metabolism are hercynine and L-ergothioneine sulfonate, both stable oxidative products of L-Erg. We did not quantify the concentration or excretion of these products, but assuming that the production of these compounds is not compatible with the formation of S-Met-L-Erg, we hypothesize that the levels of either hercynine or L-ergothioneine sulfonate or both are higher in cystinuric mice, which again would point out to a hampered oxidation regulation in cystinuric kidneys. This hypothesis is supported by the correlation observed in the levels of hercynine or L-ergothioneine sulfonate and L-Erg in kidneys [35]. The reduce glutathione levels observed in the liver of cystinuric mice might predispose to injuries caused by oxidative stress [36] and can be speculated that ergothioneine, as an antioxidant, might be being used by tissues to compensate a reduced redox metabolism.

Age-related L-Erg changes in RBCs has been observed in humans, where it declines gradually at 19–50 years from the maximum at 18 years of age [29]. We have tested L-Erg in mice at 3 and 6 months of age, which corresponds to middle age humans (20–30 years) [37]. Although a reduction in L-Erg levels were expected from human observations, the levels in RBCs at 6 months of age in comparison with those at 3 months of age were higher in mice, replicating the observations made in rats, where no age-related decline in L-Erg levels in RBCs were detected [30], suggesting specie-specific characteristics.

No gender-related differences in L-Erg plasma concentration has been reported in middle age humans [38], although it had been reported in rats [30], pointing out again to a putative specie-specific characteristic. Gender biased expression in human kidneys has been demonstrated for 67 genes [39], and 114 genes in rats [40] and glutathione metabolic process was among the functional categories of gender biased genes in human kidneys [39]. To explain the observed gender-related differences in L-Erg and S-Met-L-Erg levels in mice, it is tempting to speculate that OCTN1 expression might be gender biased in mice. We did not observed significative changes in L-Erg levels in blood, in contrast to the lower levels present in the blood of mice and patients with CKD [22], except for the increase in cystinuria male mice at 6 months of age.

Some authors have noted an increasing incidence of pediatric nephrolithiasis, at least in the U.S. [41,42], with the consequence of an increasing need of noninvasive methodologies for diagnosis and follow-up. Currently, patient’s cystine lithiasis might be controlled by analyzing the urinary crystal volume and number of cystine crystals, or the cystine capacity, or by imaging techniques. Crystalluria analysis, which is only available in selected clinical laboratories, shows an AUC of 66% [16], bellow the AUC of 84% displayed by the S-Met-L-Erg to L-Erg ratio described here. 

One of the practical limitations of crystalluria is urine processing time [16]. Such limitation could be avoided quantifying urine parameters such as cystine capacity, supersaturation, and concentration, or as we propose from our results, measuring S-Met-L-Erg and L-Erg urinary levels. The AUCs of cystine capacity, cystine supersaturation and cystine concentration are 69%, 70% and 70%, respectively [17], clearly below the 84% of the S-Met-L-Erg to L-Erg ratio. The good performance of the S-Met-L-Erg to L-Erg ratio indicates that it could be a better biomarker of cystine lithiasis than the currently existing ones.

The abovementioned methods show a limited capability to detect cystine stones in clinical settings and no reliable biomarker of cystine lithiasis can be used by clinicians to manage cystinuria patients [16]. The use of the urinary S-Met-L-Erg to L-Erg ratio in a clinical setting would not be much different of the routine amino acid determination which is nowadays mostly done by mass spectrometry [43], opening the possibility of measuring it from dry spot urine or frozen samples, once the protocol is set up in clinical diagnosis laboratories. A more reliable and easier to determine cystine lithiasis biomarker could allow clinicians to detect cystine stones earlier and prescribe appropriate treatments. An earlier detection would also mean a smaller cystine stone, leading to shorter treatment times and a lower risk of urinary tract obstruction, which could also have an impact on patients’ quality of life. Furthermore, a cystine lithiasis biomarker that does not rely on cystine could also be included in clinical assays to monitor the performance of future cystinuria treatments to prevent cystine stone formation.

The results presented here also suggest that OCTN1 (*Slc22a4*) might be a modulator of cystine lithiasis. The changes we observed in the levels of the antioxidant L-Erg together with the lower reduced and total glutathione levels described in cystinuric mice [36] and the reduction in cystine lithiasis when mice is treated with antioxidants [44,45], support the hypothesis of the involvement of redox mechanisms in cystinuria. Based on our results, it is also tempting to speculate that a lower activity or a defective OCTN1 might be behind those patients showing highly lithiasic phenotypes. 

The differences observed in S-Met-L-Erg to L-Erg ratio between lithiasic and non-lithiasic cystinuric mice and its good performance in the cystinuria mouse model support considering urinary S-Met-L-Erg to L-Erg ratio as a good biomarker of cystine lithiasis. Its use in a clinical outcome, once verified, could clearly improve cystinuria patient management and, therefore, patients’ quality of life.

## 5. Conclusions

L-ergothioneine and S-methyl-L-ergothioneine urine excretions showed significant sex-related differences in both WT and Type B cystinuria (*Slc7a9^−/−^*) mice.

In a mouse model of Type B cystinuria (*Slc7a9^−/−^*), the ratio of the urine concentrations of S-methyl-L-ergothioneine to L-ergothioneine discriminated between lithiasic and non-lithiasic mice regardless of sex and age.

The urinary S-methyl-L-ergothioneine to L-ergothioneine ratio performes as a good biomarker of cystine lithiasis in a Type B cystinuria (*Slc7a9^−/−^*) mouse model.

## 6. Patents

The results presented are part of the patent WO2021018774A1, entitled "Ergothioneine, S-methylergothioneine, and uses thereof".

## Figures and Tables

**Figure 1 antioxidants-10-01424-f001:**
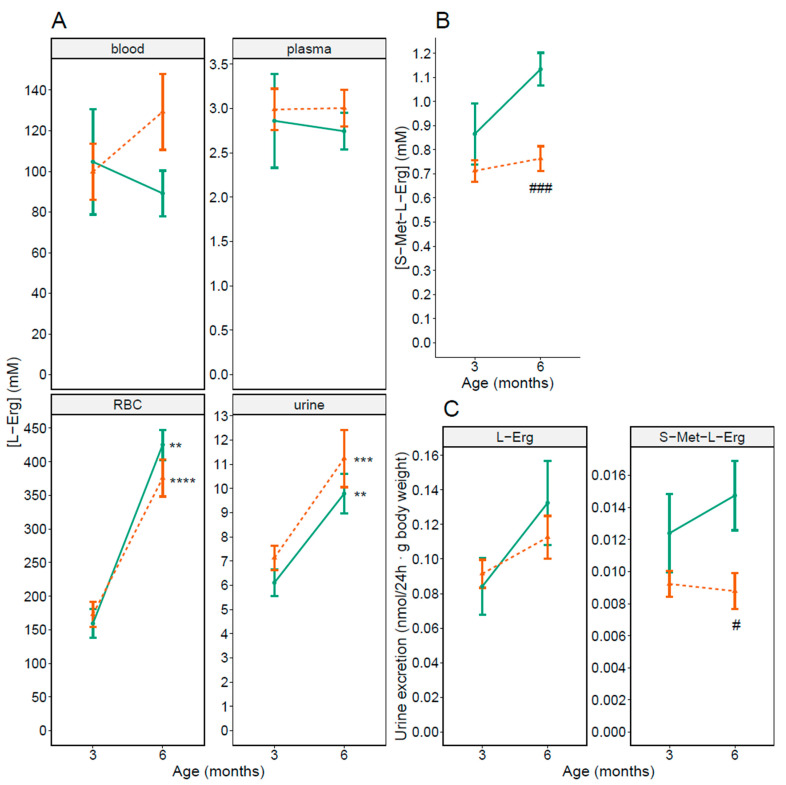
L-Erg and S-Met-L-Erg levels in WT and cystinuric (*Slc7a9^−/−^*) male mice at 3 and 6 months of age. (**A**) L-Erg concentration in blood, plasma, red blood cells (RBC) and 24 h urine in WT (green solid line) and *Slc7a9^−/−^* (orange dashed line) male mice. (**B**) S-Met-L-Erg concentration in 24 h urine. (**C**) L-Erg and S-Met-L-Erg excretion. In all panels, mean ± SEM are shown and Mann-Whitney probability test value are indicated as **, *p* ≤ 0.01; ***, *p* ≤ 0.001 and ****, *p* ≤ 0.0001 vs. 3 months of age; #, *p* ≤ 0.05 and ###, *p* ≤ 0.001 vs. WT mice. The number of independent mice used per condition ranges from 5 to 30.

**Figure 2 antioxidants-10-01424-f002:**
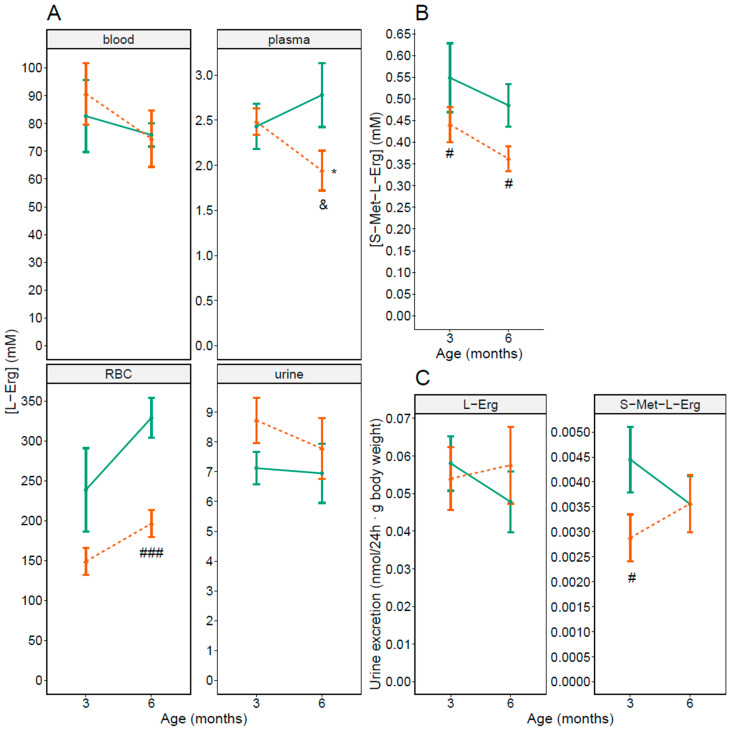
L-Erg and S-Met-L-Erg levels in WT and cystinuric (*Slc7a9^−/−^*) female mice at 3 and 6 months of age. (**A**) L-Erg concentration in blood, plasma, red blood cells (RBC) and 24 h urine in WT (green solid line) and *Slc7a9^−/−^* (orange dashed line) female mice. (**B**) S-Met-L-Erg concentration in 24 h urine. (**C**) L-Erg and S-Met-L-Erg excretion. Mean ± SEM are shown in all panels and Mann-Whitney probability test value is indicated as *, *p* ≤ 0.05 vs. 3 months of age; and &, *p* ≤ 0.1, #, *p* ≤ 0.05 and ###, *p* ≤ 0.001 vs. WT mice. The number of independent mice used per condition ranges from 6 to 31.

**Figure 3 antioxidants-10-01424-f003:**
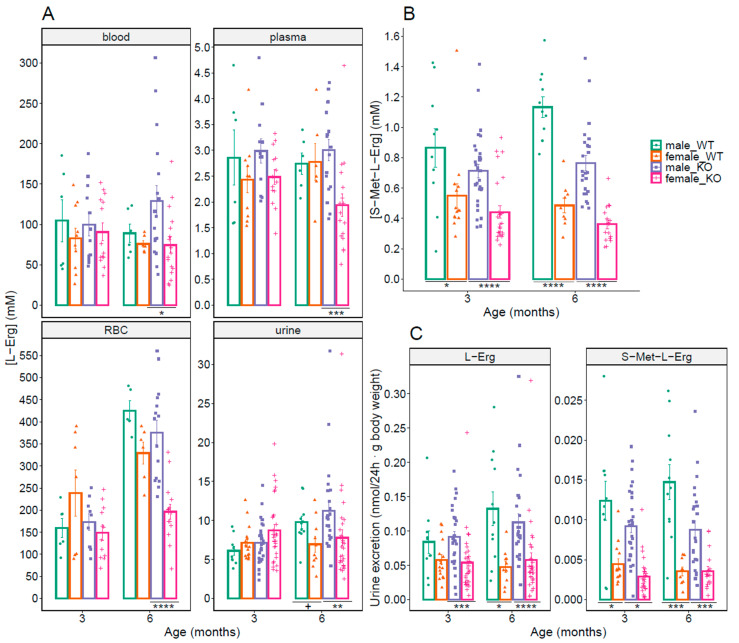
Sex-related differences of L-Erg and S-Met-L-Erg levels in WT and cystinuric (*Slc7a9^−/−^*) mice at 3 and 6 months of age. (**A**) L-Erg concentration in blood, plasma, red blood cells (RBC) and 24 h urine in WT and *Slc7a9**^−/−^* (KO) male and female mice. (**B**) S-Met-L-Erg concentration in 24 h urine. (**C**) L-Erg and S-Met-L-Erg excretion. Each colored symbol represents a sample (the number of mice per condition ranges from 5 to 31), and the bars indicate the mean ± SEM. Mann-Whitney probability test value is indicated below the bars as +, *p* ≤ 0.1; and, *, *p* ≤ 0.05, **, *p* ≤ 0.01, ***, *p* ≤ 0.001, ****, *p* ≤ 0.0001 vs. male mice.

**Figure 4 antioxidants-10-01424-f004:**
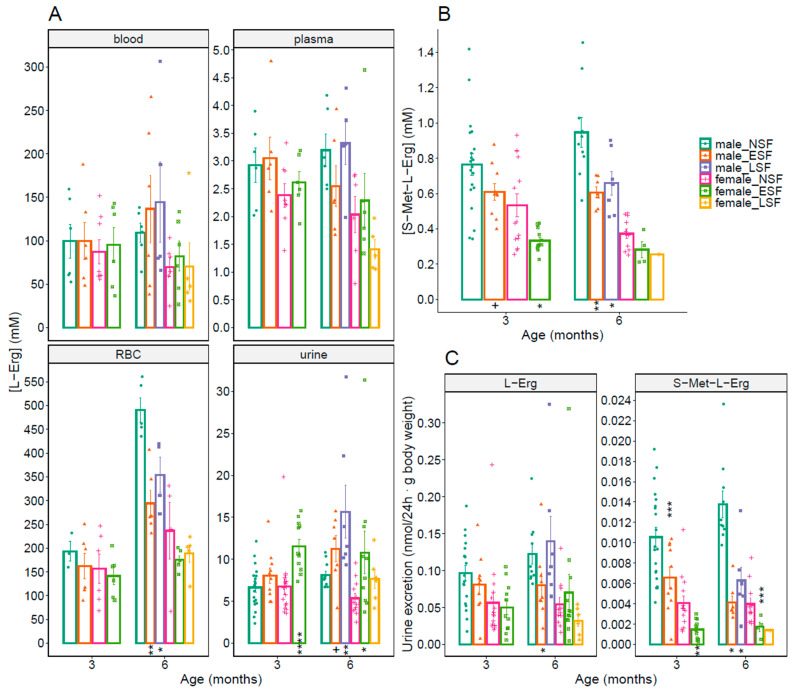
Cystine lithiasis-related differences of L-Erg and S-Met-L-Erg levels in WT and cystinuric (*Slc7a9^−/−^*) mice at 3 and 6 months of age. (**A**) L-Erg concentration in blood, plasma, red blood cells (RBC) and 24 h urine in cystine stone non-former (NSF), early stone-former (ESF) and late stone former (LSF) cystinuric mice from both sexes. (**B**) S-Met-L-Erg concentration in 24 h urine. (**C**) L-Erg and S-Met-L-Erg excretion. Each colored symbol represents a sample (the number of mice per condition ranges from 1 to 20), and the bars indicate the mean ± SEM. Mann-Whitney probability test value is indicated below the bars as +, *p* ≤ 0.1; and, *, *p* ≤ 0.05, **, *p* ≤ 0.01, ***, *p* ≤ 0.001, ****, *p* ≤ 0.0001 vs. non lithiasic (NSF) mice.

**Figure 5 antioxidants-10-01424-f005:**
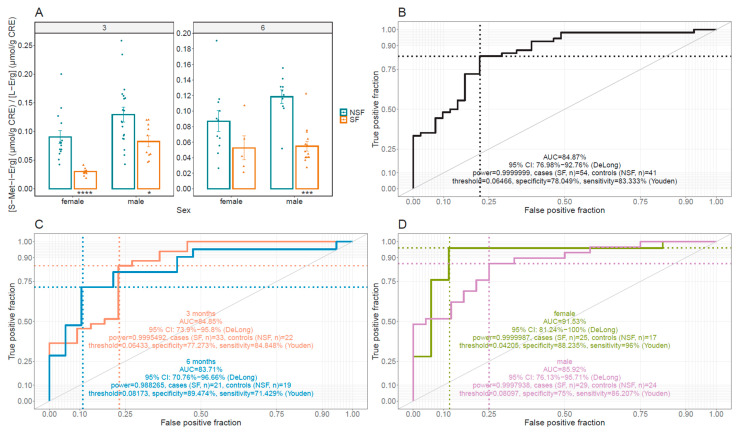
Urinary S-Met-L-Erg/L-Erg ratio in cystinuric (*Slc7a9^−/−^*) mice. (**A**) Urinary S-Met-L-Erg/L-Erg ratio in stone former (SF, orange) and non-stone former (NSF, green) *Slc7a9**^−/−^* mice stratified by age (in months) and sex. Each dot represents a sample (the number of mice per condition ranges from 5 to 19), and the bars indicate the mean ± SEM. Mann-Whitney probability test values are indicated below the bars as *, *p* ≤ 0.05, ***, *p* ≤ 0.001, ****, *p* ≤ 0.0001 vs. non-stone former mice. (**B**) Receiver operator characteristic (ROC) curve displaying the performance of the Urine S-Met-L-Erg/L-Erg ratio distinguishing SF from NSF. (**C**) Receiver operator characteristic (ROC) curve displaying the performance of the Urine S-Met-L-Erg/L-Erg ratio distinguishing SF from NSF, stratified by age. (**D**) Receiver operator characteristic (ROC) curve displaying the performance of the Urine S-Met-L-Erg/L-Erg ratio distinguishing SF from NSF, stratified by sex. In all ROC curves, AUC, Area Under the Curve, CI, Confidence Interval.

**Figure 6 antioxidants-10-01424-f006:**
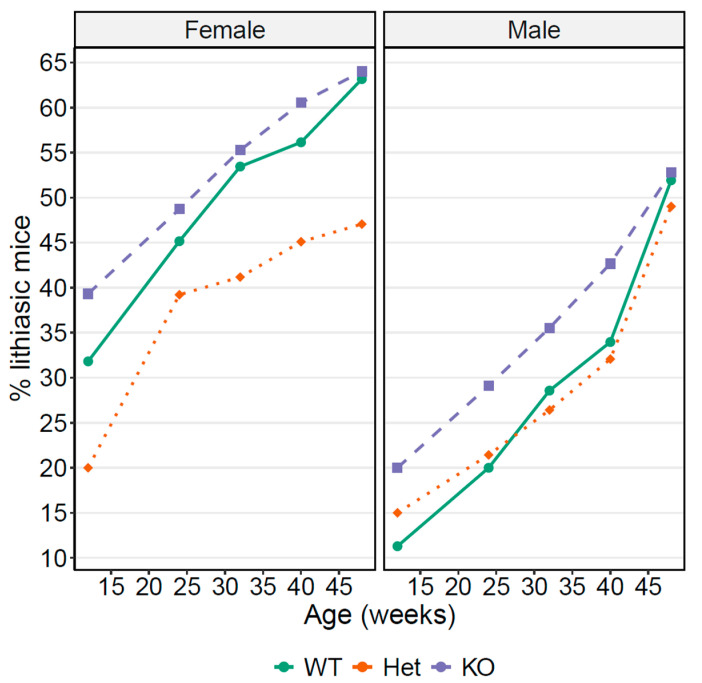
Percentage of lithiasic mice depending on *Slc22a4* genotype in a cystinuric (*Slc7a9^−/−^*) mice. The percentage of lithiasic mice in a cystinuric *Slc7a9^−/−^* genotype mice depending on *Slc22a4* genotype distributed by sex. WT mice are *Slc7a9^−/−^ Slc22a4^+/+^* (green, solid line); Het mice are *Slc7a9^−/−^ Slc22a4^+/−^* (orange dotted line); and KO mice are *Slc7a9^−/−^ Slc22a4^−/−^* (blue dashed line). Table 1 shows the number of mice per condition and age.

**Table 1 antioxidants-10-01424-t001:** Number of mice per condition and age used to assess the percentage of lithiasic mice in Figure 6.

Sex	*Slc22a4 Genotype*	*Week 12*	*Week 24*	*Week 32*	*Week 40*	*Week 48*
Female	WT (+/+)	66	62	58	57	57
Female	Het (+/−)	55	51	51	51	51
Female	KO (−/−)	89	78	76	76	75
Male	WT (+/+)	89	78	76	76	75
Male	Het (+/−)	60	60	56	53	52
Male	KO (−/−)	90	79	76	75	72

**Table 2 antioxidants-10-01424-t002:** L-Erg and S-Met-L-Erg 24 h urine concentration when knocking out *Slc22a4* in cystinuric *Slc7a9^-/-^* mice. Number of mice (N), mean and standard error (se) in µM and *p*-value. *, The *p*-value versus WT *Slc7a9* and WT *Slc22a4*, respectively. bql, bellow quantification limits. All groups contain male and female mice.

Compound	*Slc7a9* Genotype	*Slc22a4* Genotype	N	Mean	se	*p*-Value *
L-Erg	WT (+/+)	WT (+/+)	33	1.960	0.188	
L-Erg	KO (−/−)	WT (+/+)	58	1.789	0.087	0.1964
L-Erg	KO (−/−)	Het (+/−)	5	0.691	0.088	0.0382
L-Erg	KO (−/−)	KO (−/−)	3	0.176	0.028	0.0138
S-Met-L-Erg	WT (+/+)	WT (+/+)	24	0.285	0.042	
S-Met-L-Erg	KO (−/−)	WT (+/+)	56	0.152	0.013	0.0022
S-Met-L-Erg	KO (−/−)	Het (+/−)	5	bql		
S-Met-L-Erg	KO (−/−)	KO (−/−)	3	bql		

## Data Availability

Data is contained within the article.

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
