# Peer review of "S-Methyl-L-Ergothioneine to L-Ergothioneine Ratio in Urine Is a Marker of Cystine Lithiasis in a Cystinuria Mouse Model"

_antioxidants, 2021, doi:10.3390/antiox10091424_

Round 1

Reviewer 1 Report

The authors investigated urinary markers to detect or diagnose cystine stones using a mouse model of cystinuria. This is a very interesting study and is well done. I have only a few questions about this study. My concerns are:

  1. Since this study uses a mouse model, the authors need to show how well this mouse model fits into human cystinuria to avoid objections caused by species-specific properties. Can the ratio of S-methyl-L-ergothioneine to L-ergothioneine in urine be measured with as much effort as the current method, crystalluria analysis, in patients with cystinuria?
  2. Can the author argue that this method still overcomes current diagnostic methods, despite differences due to species characteristics, such as the gender differences seen in this model?

Reviewer 2 Report

Heredia et al. investigated the urinary S-methyl-L-ergothioneine to L-ergothioneine ratio in a cystinuria mouse model. It helps to discriminate between cystine lithiasis phenotypes and is considered a biomarker. I have some suggestions for this study.

1. Please provide the case numbers in each group (Slc7a9-/- male/female mouse model and WT male/female mouse model). Besides, the case numbers of L-Erg and S-Met-L-Erg levels measurement in each group are also needed.

2. The urine S-methyl-L-ergothioneine to L-ergothioneine ratio in clinical research could be discussed and strengthened the application.

3. Since the L-ergothioneine could be measured in blood, plasma, RBC, and urine, is it possible to measure circulating S-methyl-L-ergothioneine (whole blood or plasma)?

4. Urine S-Met-L-Erg / L-Erg ratio has an excellent performance in distinguishing stone former and non-stone former in overall mice. How about the receiver operator characteristic curve in male and female mice separately? Besides, urinary S-Met-L-Erg / L-Erg ratio could be presented in cystinuric mice at 3 and 6 months of age. Besides, sex stratification results of urinary S-Met-L-Erg / L-Erg ratio at 3 and 6 months of age could also be presented.

Round 2

Reviewer 2 Report

All suggestions had been revised accordingly. I have no further comments for this paper.